# A Simple and Rapid “Signal On” Fluorescent Sensor for Detecting Mercury (II) Based on the Molecular Beacon Aptamer

**DOI:** 10.3390/foods11131847

**Published:** 2022-06-23

**Authors:** Li Wang, En-Zhong Chi, Xin-Huai Zhao, Qiang Zhang

**Affiliations:** 1School of Biology and Food Engineering, Guangdong University of Petrochemical Technology, Maoming 525000, China; wangli7742@gdupt.edu.cn (L.W.); junfengchi@gdupt.edu.cn (E.-Z.C.); zhaoxh@gdupt.edu.cn (X.-H.Z.); 2Engineering Research Center for Food Safety and Nutritional Evaluation, Maoming 525000, China

**Keywords:** aptamer, molecular beacon, mercury (II), detection method

## Abstract

Biosensors for mercury (II) (Hg^2+^) with high sensitivity are urgently required for food safety, ecosystem protection and disease prevention. In this study, a simple and fast detection method of Hg^2+^ based on the molecular beacon aptamer was established, according to the principle that Hg^2+^ could change the structure of the molecular beacon aptamer, resulting in the changed fluorescence intensity. All of the detection conditions were optimized. It was found that an optimal molecular beacon aptamer MB3 showed the optimal response signal in the optimized reaction environment, which was 0.08 μmol/L MB3, 50 mmol/L tris buffer (40 mmol/L NaCl, 10 mmol/L MgCl_2_, pH 8.1), and a 10 min reaction. Under the optimal detection conditions, the molecular beacon aptamer sensor showed a linear response to Hg^2+^ concentration within a range from 0.4 to 10 μmol/L and with a detection limit of 0.2254 μmol/L and a precision of 4.9%. The recovery rates of Hg^2+^ in water samples ranged from 95.00% to 99.25%. The method was convenient and rapid, which could realize the rapid detection of mercury ions in water samples.

## 1. Introduction

Mercury is one of the most toxic heavy metals with well documented impacts to the environment and human health [1,2], with strong volatility, low melting point and high toxicity. Mercury and its compounds (Hg^2+^) mainly enter the human body through the respiratory tract, skin, digestive tract and other channels. Low dose and long-term exposure will cause chronic damage to human organs, such as the liver, kidney, reproductive system and nervous system [3]. In recent years, due to the rapid development of the chemical industry, there are numerous cases of mercury poisoning. Therefore, the monitoring of mercury in water and food is crucial. There are various methods used for the detection of Hg^2+^, including atomic absorption spectroscopy (AAS) [4,5], atomic fluorescence spectrometry (AFS) [6], inductively coupled plasma mass spectrometry (ICP-MS) [7], surface-enhanced Raman spectroscopy (SERS) [8], etc. However, these methods need professional operation, a special testing machine and relatively tedious steps. These greatly affect the timeliness of field detection of Hg^2+^. It is important to develop rapid, convenient and cost-efficient detection methods for Hg^2+^.

Aptamers are oligonucleotides that can specifically bind to proteins or other small molecules with high affinity and specificity. They are selected using an in vitro selection procedure called SELEX, the systematic evolution of ligands with exponential enrichment [9,10]. After decades of development, aptamers have been widely used in many fields, such as sensors and molecular biology. Because the thymine-thymine (T-T) mismatches in DNA duplexes can selectively and tightly bind to Hg^2+^, the T-rich oligonucleotide has been frequently applied as a Hg^2+^ specific aptamer (MSA) [11,12].

A molecular beacon is a stem-ring double-labeled oligonucleotide probe that forms a hairpin structure at the 5′and 3′ ends. The loop region contains an antisense recognition domain against a nucleic acid target, and the absence of a target allows the molecular beacon to self-hybridize at the stem [13]. Since the development of the first molecular beacon by Tyagi and Kramer [14] in 1996, many studies have demonstrated their use as sensors. It has been widely applied in chemistry, biology, medicine, and other fields, especially in the detection of protein and polypeptide, real-time imaging of living cells, etc [15].

Generally, a molecular beacon probe contains 25–35 nucleotides, and its structure includes loop region, stem region and labeling region. The loop region is composed of 15 to 30 nucleotides, which can specifically bind to the target molecule. The stem region is composed of five to eight base pairs, which can be reversibly dissociated when the molecular beacon binds to the target molecule. In the labeling region, the fluorescent group is generally labeled at the 5′ end, and the quenching group is generally labeled at the 3′ end. In the free state, the molecular beacon has a hairpin structure, and the fluorescence groups and quenching groups are close to each other; at this time, the fluorescence resonance energy transfer occurs, and the fluorescence is almost completely quenched. When the molecular beacon binds to the target molecule, its spatial configuration changes, the distance between the fluorescence group and the quenching group increases, and the fluorescence of the molecular beacon is restored. Compared with other types of nucleic acid probes (such as constructing DNA double-stranded hybridization system), the structure of the molecular beacon is simple, and one kind of molecular beacon can detect one kind of target in a homogeneous system. At the same time, it has the characteristics of low background fluorescence, simple operation, high sensitivity and strong specificity.

A molecular beacon aptamer combines the advantages of an aptamer and molecular beacon organically. In this paper, a molecular beacon aptamer was constructed and used for Hg^2+^ detection. The fluorescent group was labeled at the 5′ end and the quenching group was labeled at the 3′ end. When Hg^2+^ was added, the T base in the molecular beacon aptamers formed the structure of T-Hg^2+^-T. The specific binding of T base and Hg^2+^ led to the conformational change in the molecular beacon aptamer, which enlarged the distance between the fluorescence group and quenching group, so the fluorescence intensity increased. Based on this principle, a rapid detection method for Hg^2+^ was established, which developed a new way for the detection of Hg^2+^.

## 2. Materials and Methods

### 2.1. Instruments and Reagents

A FilterMax F3 multi-mode microplate reader (Molecular Devices, USA), 96-well, black microplate (Costar, Washington, DC, USA), PHS-3B pH meter (Hongyi Instrumentation Co. Ltd., Shanghai, China) and a TGL-16 high-speed refrigerated centrifuge (Xiangyi Lab Instrument Co. Ltd., Hunan, China) were used.

Tris, NaCl, MgCl_2_, HCl, HgCl_2_, ZnCl_2_, AlCl_3_, MnCl_2_, SnCl_2_, CoCl_2_, KCl, CuCl_2_, BaCl_2_, CaCl_2_ and CdCl_2_ were used as the analytical reagents. The water used in the experiment was ultra-pure water.

### 2.2. DNA Sequence of the Molecular Beacon Aptamer

The DNA sequences used in the experiment were all synthesized by Shenggong Bioengineering (Shanghai) Co., Ltd. and the following DNA sequences were used: MB1, GAACACCCTTCTTCTTCCTTGTTGTTC (5′FAM, 3′DABCYL); MB2, GAACACCCCTTC TTCTTCCTTGTTGTTC (5′FAM, 3′DABCYL); MB3, GAACACCCCCTTCTTCTTCCTTGT TGTTC (5′FAM, 3′DABCYL).

### 2.3. Selection of the Molecular Beacon Aptamer

In the buffer (50 mmol/L tris, 50 mmol/L NaCl, 10 mmol/LMgCl_2_, pH 7.2), 50 μL of 0.1 μmol/L molecular beacon aptamer was added to 50 μL of 50 μmol/L Hg^2+^ solution (the sample group) or 50 μL buffer (the blank group). After they were mixed, the fluorescence intensity was detected every five minutes by a multi-function microwell plate detector in a black 96-well plate, with an excitation wavelength 485 nm and emission wavelength of 535 nm.

### 2.4. Selection of MB3 Concentration and Reaction Time

The MB3 with the best sensitivity to Hg^2+^ was selected by Section 2.3. Then, 50 μL of MB3 at different concentrations (0.02, 0.04, 0.06, 0.08 and 0.1 μmol/L) were added to 50 μmol/L Hg^2+^ solution (the sample group) or 50 μL buffer (the blank group). Afterwards, the fluorescence intensity was detected according to the method in Section 2.3. *I* and *I_0_* were the fluorescence intensity of the sample group and the blank group, respectively, and the fluorescence intensity change ∆*I* (∆*I *=* I − I_0_*) was calculated.

### 2.5. Selection of pH and Ionic Strength in the Buffer

After the optimal molecular beacon aptamer concentration and the optimum reaction time were determined, the effects of different pH (pH 7.2, 7.5, 7.8, 8.1 and 8.4), NaCl concentration (30, 40, 50, 60 and 70 mmol/L) and MgCl_2_ concentration (5, 10, 15, 20 and 25 mmol/L) in the buffer were studied. Under the optimal reaction and test conditions, the fluorescence intensity was detected and the ∆*I* was calculated according to the method in Section 2.4.

### 2.6. Specificity Experiment

In the blank group, 50 μL of optimum concentration MB3 and 50 μL of optimized buffer solution were added. In the sample group, 50 μL of optimum concentration MB3 and 50 μL of 50 μmol/L Zn^2+^, Al^3+^, Mn^2+^, Sn^2+^, Co^2+^, K^+^, Cu^2+^, Ba^2+^, Ca^2+^, Cd^2+^ and Hg^2+^ in the optimized buffer solution were added. The detection and calculated methods were the same as Section 2.5.

### 2.7. Affinity Experiment and Standard Curve

The blank group experiment was the same as Section 2.6. In the sample group, 50 μL of MB3 at the optimal concentration and 50 μL of Hg^2+^ at different concentrations were added. The ∆*I* was calculated, and the relationship between the concentration of Hg^2+^ and the ∆*I* was established. The *Kd* of MB3 was fitted and calculated by GraphPad Prism 5 software.

### 2.8. Detection Limit and Precision

According to the Operational Guidelines for Water Sample Detection of Global Environmental Monitoring System, when the confidence level was 95%, the detection limit was the one-time measured value of the sample concentration, which was significantly different from the one-time measured value of the zero concentration sample. When the number of blank measurements was greater than 20, the detection limit was 4.6 times that of the blank standard deviation. The precision of the method was evaluated by the relative standard deviation of five repeated tests.

### 2.9. Recovery Test of Hg^2+^ in Water Samples

Water from tap water was collected and filtered with a 0.22 μm filter membrane, and the initial concentration of Hg^2+^ was detected. The 0.8, 4 and 8 μmol/L Hg^2+^ solutions were added to the filtered water samples. Then, the samples were tested according to the operation of Section 2.7. The recovery rate was calculated according to the following formula:*P* = (*C*_2_ – *C*_1_)/*C*_3_ × 100%

*P* was the recovery rate, *C*_1_ was the initial concentration of Hg^2+^ in the water, *C*_2_ was the determination of Hg^2+^ concentration in the standard addition group, and *C*_3_ was the standard adding concentration of Hg^2+^.

## 3. Results

### 3.1. Detection Principle

As shown in Figure 1, the molecular beacon aptamer in the experiment was a hairpin structure, so the 5′-end fluorescent group (FAM) and the 3′-end quenching group (DABCYL) were very close to each other, and it was easy to generate fluorescence resonance energy transfer. So, the fluorescence of the fluorescent group was quenched and the fluorescence intensity was low. When Hg^2+^ was added, the T-base and Hg^2+^ formed the structure of T-Hg^2+^-T. This led to the changed structure of the molecular beacon aptamer, the distance between the fluorescent group and quenching group became longer, and the fluorescence intensity increased greatly. The concentration of Hg^2+^ was directly proportional to the change in fluorescence intensity of the molecular beacon aptamer, so the concentration of Hg^2+^ could be detected.

### 3.2. Design and Verification of Molecular Beacon Aptamer

Three molecular beacon aptamers, MB1, MB2 and MB3, were designed in the experiment, and the MFold webserver was used to predict their secondary structures. As shown in Figure 2, each molecular beacon aptamer had a hairpin structure and had five base pairs in the neck. Theoretically, the FAM fluorescent group at the end of 5′ and the DABCYL quenching group at the end of 3′ were very close, resulting in fluorescence resonance energy transfer and low fluorescence intensity.

The experiment investigated whether the molecular beacon aptamer can be successfully closed and opened by detecting the fluorescence intensity of the system before and after adding Hg^2+^. The results were shown in Figure 3. In the absence of mercury ions, the fluorescence intensities of MB1, MB2 and MB3 were 9576.125, 7149.125 and 6942.750 at 10 min, respectively. However, the fluorescence intensities increased significantly after the addition of 50 μmol/L Hg^2+^, the fluorescence intensities increased to 22,549.000, 26,630.130, and 28,116.130, respectively, and the fluorescence intensity after adding Hg^2+^ was 2 to 4 times higher than that without Hg^2+^.

This proved that three molecular beacon aptamers were able to bind to Hg^2+^ smoothly. Among them, the fluorescence intensity change in MB3 was the highest, so MB3 was chosen as the optimal molecular beacon aptamer.

### 3.3. MB3 Concentration and Incubation Time

The changes in fluorescence intensity of the different concentrations of MB3 reacting with 50 μmol/L Hg^2+^ were detected. As can be observed from Figure 4, with the increase in the concentration of MB3, Δ*I* was increased, and it was the highest at 0.08 μmol/L. When the concentration of MB3 was further increased, the change value of fluorescence intensity was decreased. When MB3 concentration was 0.08 μmol/L, with the extension of reaction time, Δ*I* showed a slight increase at first and then gradually decreased, and reached the maximum value at 10 min.

### 3.4. pH and Ionic Strength in the Buffer

Under the conditions that the concentration of Hg^2+^ was 50 μmol/L, the concentration of MB3 was 0.08 μmol/L, and the incubation time was 10 min, the effects of buffer pH (tris 50 mmol/L, 50 mmol/L NaCl, 10 mmol/L MgCl_2_) on the detection system were studied. The results (Figure 5A) showed that the pH of the buffer had a great influence on the Δ*I* between pH 7.2 and pH 8.4, and the most remarkable change in fluorescence intensity was at pH 8.1.

Under the conditions that the concentration of Hg^2+^ was 50 μmol/L, the concentration of MB3 was 0.08 μmol/L, and the incubation time was 10 min, the effects of different concentrations of NaCl (tris 50 mmol/L, 10 mmol/L MgCl_2_, pH 8.1) on the fluorescence intensity of the detection system were studied. The results (Figure 5B) showed that Δ*I* increased first and then decreased when the concentration of NaCl was 30–70 mmol/L, and the change value of fluorescence intensity was the most significant when the concentration of NaCl was 40 mmol/L. According to the same method, the concentration of MgCl_2_ changed in the range of 5–25 mmol/L (tris 50 mmol/L, 40 mmol/L NaCl, pH 8.1); the results were shown in Figure 5C. When the concentration of MgCl_2_ reached 10 mmol/L, the Δ*I* was the highest.

### 3.5. Specificity Test

Under the optimal reaction conditions, Δ*I* of MB3 interacting with different metal ions was detected, and the results are shown in Figure 6. It was found that among the 11 metal ions, *ΔI* of MB3 reacting with Hg^2+^ was the highest (Δ*I* = 47161.375), while those of other metal ions were relatively low or even had no change. Among the other ten kinds of metal ions, Ba^2+^, Ca^2+^, Cd^2+^, Al^3+^ and Co^2+^ had a relatively large influence on the detection, and the *ΔI* was 2732.875, 1834.500, 1546.125, −1356.500 and −780.200, respectively. However, their *ΔI* accounted for less than 5.79% of the Δ*I* of Hg^2+^. It was proved that the molecular beacon aptamer MB3 had excellent specificity.

### 3.6. Affinity Test, Standard Curve, Detection Limit and Precision

Under the optimal conditions, Δ*I* of MB3 reacting with 0.4–50 μmol/L Hg^2+^ was detected, and the results are shown in Figure 7A. In the range of Hg^2+^ concentration from 0.4 to 10 μmol/L, the change in fluorescence intensity increased with the increase in Hg^2+^ concentration. However, when the Hg^2+^ was higher than 10 μmol/L, Δ*I* tended to be stable. The *K_d_* of MB3 was fitted and calculated by GraphPad Prism 5 software. It can be observed from Figure 7 that the *K_d_* of the MB3 was 8.567 ± 1.558 μmol/L.

In the range from 0.4 to 10 μmol/L, the Δ*I* had a linear relationship with the concentration of Hg^2+^. The standard curve was y = 3506.6x − 960.57; the linear regression coefficient was 0.9901 (Figure 7B). The detection limit of the present method was 0.2254 μmol/L, and the relative standard deviation of the five repeated tests was 4.9%.

### 3.7. Recovery Test of Hg^2+^ in Water Samples

Using the established detection method, the standard addition and recovery test of the water samples was carried out. It can be observed from Table 1 that the recovery rate of Hg^2+^ in the water sample was between 95.00% and 99.25%. The accuracy rate was high, which could be used for actual sample determination.

## 4. Discussion

Aptamers have received more and more attention in the construction of biosensors in the past few years. Recently, various strategies based on aptamers for metal ion detection have been developed, including Hg^2+^ [16,17], Pb^2+^ [18], Cd^2+^ [19] and so on. Studies had shown that thymine-thymine (T-T) mismatched duplexes in DNA can selectively bind closely to Hg^2+^ to form thymine-Hg^2+^-thymine (T-Hg^2+^-T)-mediated DNA duplexes [12]. Based on this characteristic, T-rich oligonucleotides have been used as specific aptamers for Hg^2+^ analysis. The constructed aptamer biosensors for Hg^2+^ detection include colorimetric sensors [3,20], fluorescence sensors [1,21,22], electrochemical sensors [12,23] and so on. Most of the biosensors have been combined with a variety of nanomaterials, such as nano-gold, nano-silver, nanowires, carbon nanotubes, graphene, quantum dots, fluorescent metal nanoclusters, and other materials [16,24,25], and the detection process is relatively cumbersome.

In this study, a simple and convenient “signal on” fluorescence sensor was constructed. The advantages of an aptamer and molecular beacon were organically combined, and the molecular beacon aptamer for Hg^2+^ detection was designed. In the detection system, there was only one type of nucleic acid probe, and there was no need to hybridize various types of the nucleic acid probe, so the detection system and operation are relatively simple. The results showed that Hg^2+^ could induce the conformational change in the molecular beacon aptamer by forming a T-Hg^2+^-T structure, resulting in the fluorescence signal changes to realize the detection of Hg^2+^. The MB3 probe designed in this study had strong affinity, and the *K_d_* of the MB3 was 8.567 ± 1.558 μmol/L. The study of Qu. et al. [26] found that the *K_d_* of the selected aptamer binding to Hg^2+^ was 1.49 ± 1.558 μmol/L, and the *K_d_* of the previously reported Hg^2^^+^ aptamer [11] was 34.08 ± 4.07 μmol/L. The *Kd* of the MB3 probe was between 1.49 ± 1.558 μmol/L and 34.08 ± 4.07 μmol/L. This could also prove that the designed molecular beacon was successful and effective.

In addition, this method has good specificity, precision and stability. Among the other 10 kinds of metal ions tested, Ba^2+^, Ca^2+^, Cd^2+^, Al^3+^ and Co^2+^ had a certain influence on the detection, but the overall influence was small. When the method was used for detection, the relative standard deviation of five repeated tests was 5.9%, which indicates that the precision of the method is good. At the same time, it was found that the fluorescence intensity of the reaction system changed very little and the detection was stable when the reaction time was from 10 to 35 min. The detection limit of the method was 0.2254 μmol/L.

In the study of Wang et al. [3], a colorimetric assay for the determination of Hg^2+^ was demonstrated with unmodified gold nanoparticles as probes and thrombin-binding aptamer as sensing elements; the limit of detection was 200 nmol/L. This result was similar to that of our study. However, in the study of Ono et al. [11], a “signal off” fluorescent sensor was used to detect Hg^2+^; the limit of detection was 40 nmol/L. The detection limit is lower than ours, so our detection sensitivity needs to be improved and optimized.

In addition, the reaction conditions of the experiment, including the concentration of MB3, reaction time, pH and ionic strength of the buffer, were comprehensively optimized. Judging from the results, the higher concentration of MB3 was not the best. A higher concentration increased the probability of MB3 binding to Hg^2+^, but the background value of MB3 fluorescence intensity also increased, resulting in the decreased ΔI. In terms of the reaction time, the reaction speed was very fast after adding Hg^2+^, and the fluorescence intensity tended to be stable at about 10 min, which laid the foundation for rapid detection. From pH and ion strength of the buffer, the weak alkaline condition and a certain concentration of sodium ion and magnesium ion were beneficial to the binding of the molecular beacon aptamer to Hg^2+^ and detection. Especially, the pH of the buffer system had a great influence on the results; the optimized pH in this study was 8.1, which was close to the pH of buffer system reported by previous studies. For example, Li et al. [20] used a buffer of pH 8.0 when they detected mercury ions based on unmodified gold nanoparticles and aptamers. Similarly, Tan et al. [27] used an aptamer functionalized gold nanoparticle fluorescent probe to detect mercury (II) in an aqueous solution, and the pH of the buffer was 8.2.

Overall, the method has the advantages of fewer reagents, simple operation, rapid detection and low cost. It can complete the detection of mercury ions in water samples within 20 min, showing good specificity and certain sensitivity. It can be used as a rapid screening method for Hg^2+^, but its detection sensitivity needs to be improved and optimized. The study focuses on the design of a nucleic acid probe and the establishment of a detection method, which is only preliminarily applied in tap water. However, its application in other samples needs to be further studied.

## 5. Conclusions

A simple and rapid “signal on” fluorescent sensor for detecting Hg^2+^ based on a molecular beacon aptamer was developed, which was used for the detection of water samples. The designed molecular beacon aptamer could specifically and closely bind to Hg^2+^ to form a thymine-Hg^2+^-thymine (T-Hg^2+^-T)-mediated structure, which changed the hairpin structure of the original molecular beacon aptamer and increased the fluorescence intensity of the system. The MB3 probe designed in this study had high affinity and specificity. This method had a good linear relationship in the range of 0.4–10 μmol/L. The method had a certain anti-interference ability, and the recovery rate in the water sample was over 95.00%. The method did not require a complicated sample pretreatment process, the detection system was simple, and the detection could be completed within 20 min, which paves the way for the rapid screening of Hg^2+^.

## Figures and Tables

**Figure 1 foods-11-01847-f001:**
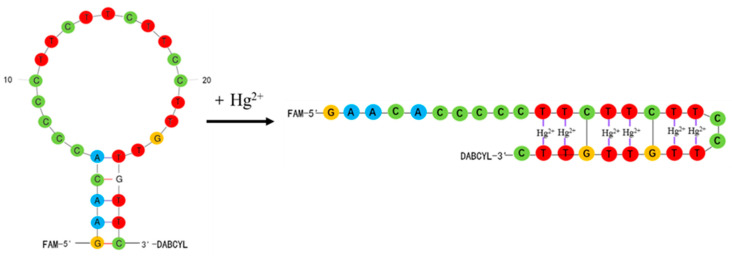
Principle of the molecular beacon aptamer detecting Hg^2+^.

**Figure 2 foods-11-01847-f002:**
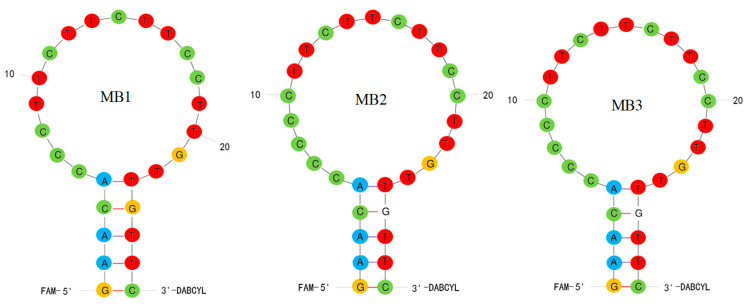
Secondary structures of three molecular beacon aptamers.

**Figure 3 foods-11-01847-f003:**
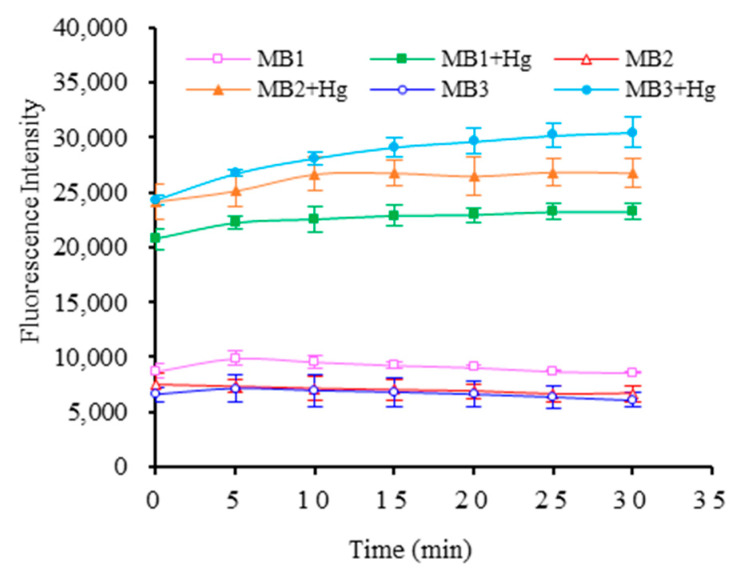
Fluorescence intensity of three molecular beacon aptamers responding to Hg^2+^.

**Figure 4 foods-11-01847-f004:**
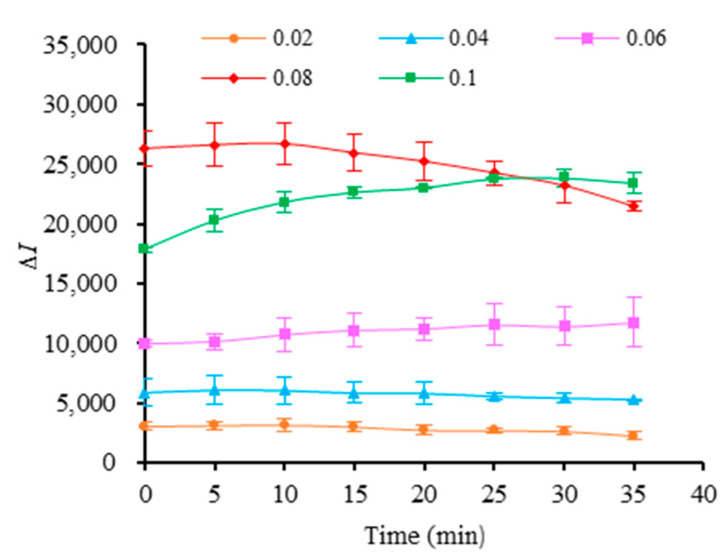
The fluorescence intensity changes in different concentrations of MB3 reacting with Hg^2+^.

**Figure 5 foods-11-01847-f005:**
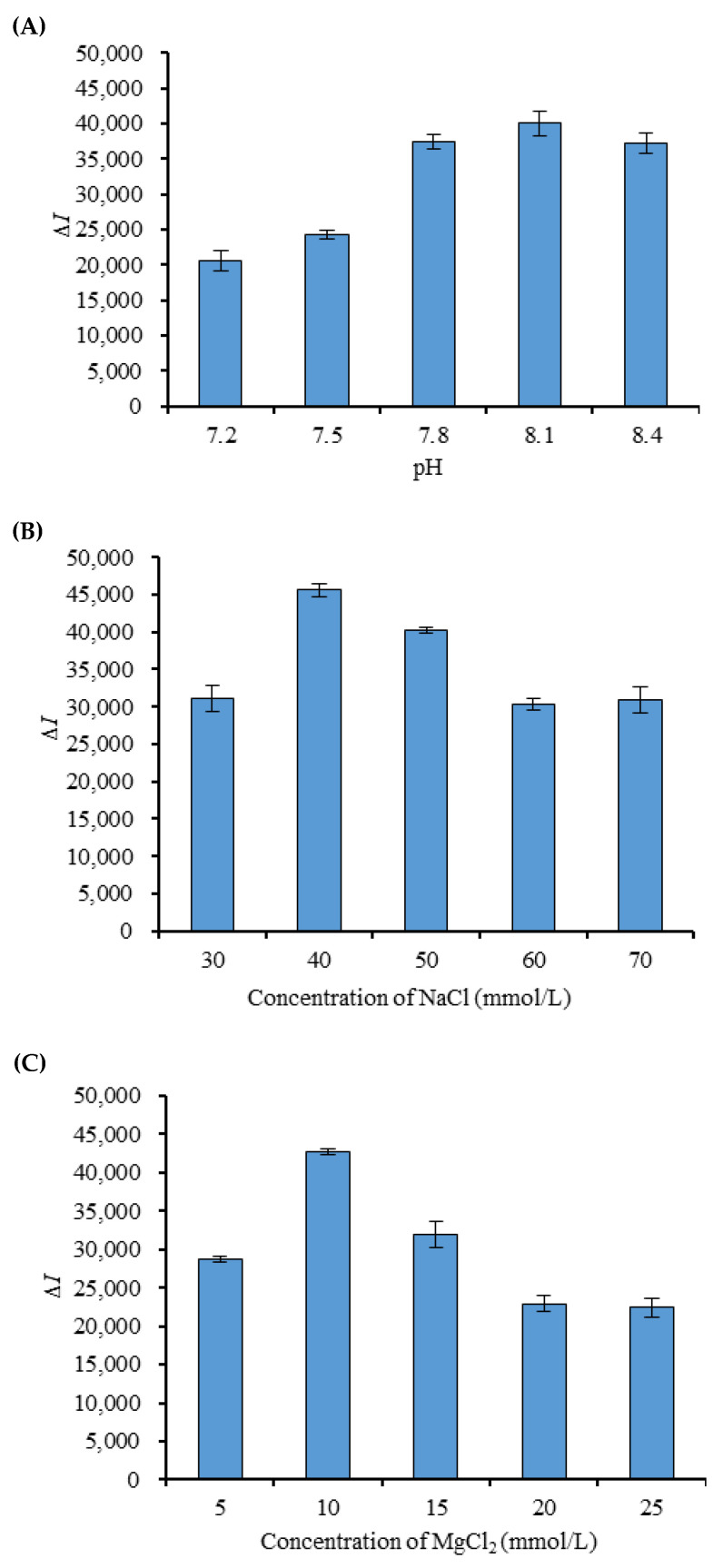
The effects of pH (**A**), concentrations of NaCl (**B**) and concentrations of MgCl_2_ (**C**) on the fluorescence intensity.

**Figure 6 foods-11-01847-f006:**
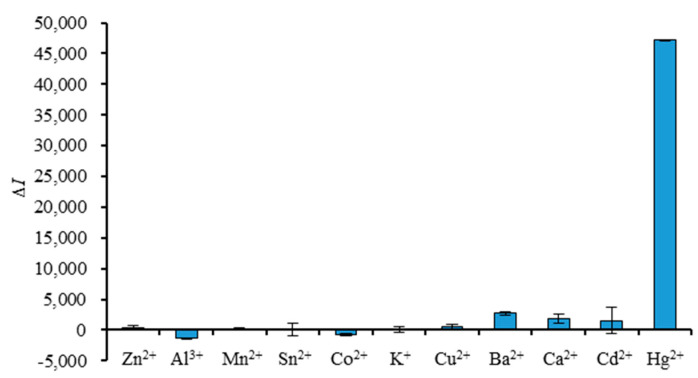
The fluorescence intensity changes in MB3 interacting with different metal ions.

**Figure 7 foods-11-01847-f007:**
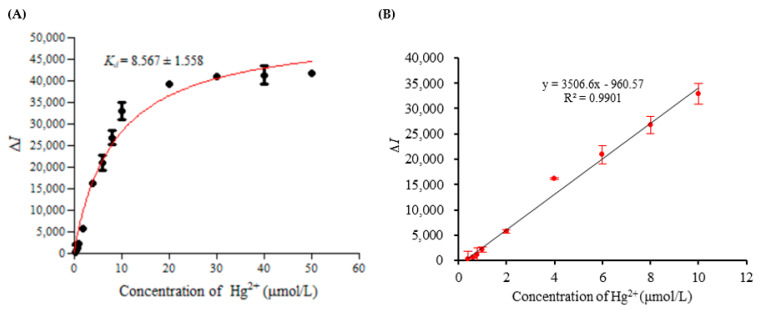
(**A**) The relationship between the fluorescence intensity changes and Hg^2+^ concentrations. (**B**) The standard curve.

**Table 1 foods-11-01847-t001:** Recovery test of Hg^2+^ in water samples.

Initial Concentration of Hg^2+^ (μmol/L)	Standard Adding Concentration of Hg^2+^ (μmol /L)	Determination of Hg^2+^ Concentration (μmol/L)	Recovery Rate (%)
Not detected	8	7.90 ± 0.5	98.75
Not detected	4	3.97 ± 0.3	99.25
Not detected	0.8	0.76 ± 0.1	95.00

## Data Availability

All data are contained within the article.

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
