# Peer review of "A Simple and Rapid “Signal On” Fluorescent Sensor for Detecting Mercury (II) Based on the Molecular Beacon Aptamer"

_foods, 2022, doi:10.3390/foods11131847_

Round 1
Reviewer 1 Report
a rapid detection method of tracing Hg(II) based on the molecular beacon aptamer, according to the principle that Hg(II) could change the structure of the molecular beacon aptamer resulting in a change in the fluorescence intensity. All the detection conditions were investigated. It is clear that an optimal molecular beacon aptamer MB3 showed the optimal response signal in the optimized reaction environment, which was 0.08 mmol/L MB3, 50 mmol/L Tris buffer (40 mmol/L NaCl, 10 mmol/L MgCl2, pH 8.1), and 10 min reaction. Under the optimal detection conditions, the molecular beacon aptamer sensor showed a linear response to Hg(II) concentration within a range from 0.4 to 10 mmol/L and with a detection limit of 0.2254 mmol/L and a precision of 4.9%. The recovery rates of Hg(II) in water samples ranged from 95.00% to 99.25%. The method was convenient and rapid, which could realize the rapid detection of mercury ions in water samples. Therefore, I recommend this manuscript for publication in molecules after major revision.
Introduction:
- Insert a new paragraph to explain the advantages of the Molecular Beacon sensing technique with respect to other utilizing techniques?
- Clarify the benefits of (Molecular Beacon) make it the desired choice than others?
- provide short notes about the applied fluorescence mechanism and compare this mechanism with other utilized mechanisms?
Results and discussion
- The advantages of this method in comparison with other methods should be highlighted, including analytical characteristics, reproducibility, specificity, and stability?
- No data about reversibility are given (in the presence of Hg(II)?
This is a critical issue!
- Apply the Bensi-Hildebrand equation to calculate the binding constant?
- The validation of this technique should be introduced by comparison with a previously validated method.
- The effect of the interfering ions on the detection efficiency in the presence of Hg(II) ions
- The conclusion should be containing more details
Reviewer 2 Report
The authors report a Simple and Rapid "Signal On" Fluorescent Sensor for detect Mercury (II) Based on the Molecular Beacon Aptamer. I think the manuscript is suitable for the publication in this journal after considering my following concerns:
* The luminescence lifetime of this material should be provided in this manuscript.
* According to this manuscript, this material can be used for detection of Hg2 in an aqueous medium. However, can it also be applied to the gas detection of mercury? This should be discussed more, at least with one paragraph.
* Several relevant references about applications of fluorescent compounds can be cited, such as Chemical Engineering Journal, 2022, 432, 134327 (DOI: 10.1016/j.cej.2021.134327); Inorg. Chem., 2022, 61, 2883-2891 (DOI: 10.1021/acs.inorgchem.1c03563); Journal of Luminescence 243, 2022, 118668 (DOI 10.1016/j.jlumin.2021.118668)
* The widespread use of Hg2+ cations has caused serious pollution to the environment. Therefore, it would be better to include tests for other water samples (river water or seawater) in the text.
* comparison of the limit of detection between the presented material and other probes reported in the literature should be added in the text
Reviewer 3 Report
A comprehensive study of molecular beacon aptamer as "Signal On" fluorescent sensor for determination of mercury(II) cations in water samples was conducted. Sensor demonstrate high selectivity and rapid determination of mercury(II) cations up to 0.2254 μmol/L. However, it is not clear for me the determination of mercury(II) cations in the presence of interfering ions has been investigated or interaction with other cations was studied only for each ion separately as shown in Figure 8. The cross-selectivity experiment is important due to the fact, that presence of interfering ions can prevent the binding of mercury(II) cations by sensor.
Round 2
Reviewer 1 Report
Authors reply significantly to all required comments. Thus, this research is well arranged, with a sequence of clear ideas and concise writing that fits the research plan and methodology. The literature review is good, and they were able to successfully discuss their progress from both a perspective and an applied perspective. Their chosen method makes this data analysis excellent research and enables them to answer research questions and test their hypotheses.
Reviewer 2 Report
None